# DIFER: Differentiable Automated Feature Engineering

Guanghui Zhu[1]  Zhuoer Xu[1]  Chunfeng Yuan[1]  Yihua Huang[1]

[1]State Key Laboratory for Novel Software Technology, Nanjing University

**Abstract**  Feature engineering, a crucial step of machine learning, aims to construct useful features from raw data to improve model performance. In recent years, great efforts have been devoted to Automated Feature Engineering (AutoFE) to replace expensive human labor. However, all existing methods treat AutoFE as an optimization problem over a discrete feature space, leading to the problems of feature explosion and computational inefficiency. Unlike previous work, we perform AutoFE in a continuous vector space and propose a differentiable method called DIFER in this paper. Specifically, we first propose an evolutionary framework to search for better features iteratively. In each feature evolution step, we introduce a feature optimizer based on the encoder-predictor-decoder, which maps features into the continuous vector space via the encoder, optimizes the embedding along the gradient direction induced by the predictor, and recovers better features from the optimized embedding by the decoder. Extensive experiments on classification and regression datasets demonstrate that DIFER can significantly outperform the state-of-the-art AutoFE method in terms of both model performance and computational efficiency. The implementation of DIFER is avaialable on `https://github.com/PasaLab/DIFER`.

## 1 Introduction

Feature engineering, the process of constructing features from raw data, directly determines the upper bound of various machine learning algorithms (e.g., Random Forest and Logistic Regression). However, it requires considerable domain knowledge to construct features. Also, huge computational resources are needed to evaluate and then filter features. Thus, it is a cost-intensive task to find useful and meaningful features.

Recently, the AutoFE (Automated Feature Engineering) methods that search for useful features without any human intervention have received more and more attention. AutoFE formalizes feature construction as applying transformations (e.g., arithmetic operators) to the raw features. The *expansion-reduction* algorithm (Kanter and Veeramachaneni, 2015; Lam et al., 2017) iteratively applies all transformations to each feature and selects the features based on the model performance. Without expert guidance, such method consumes significant computational resources for feature evaluation due to the exponentially growing feature space. To reduce the cost, learning-based AutoFE methods are proposed. TransGraph (Khurana et al., 2018) trains a Q-learning agent to decide the transformation. Due to applying each action (i.e., transformation) to all features, TransGraph also suffers from the feature explosion problem. LFE (Nargesian et al., 2017) trains an MLP (Multi-Layer Perceptron) to recommend the most likely useful transformation for each feature. However, it does not support the composition of transformations. NFS (Chen et al., 2019) generates a feature transformation sequence for each raw feature under the guidance of an RNN controller. Although NFS can achieve SOTA (state-of-the-art) performance, the computational efficiency is still low. An inherent cause of inefficiency for the existing approaches is the fact that AutoFE is treated as an optimization problem over a discrete space.

In this paper, we address the AutoFE problem from a different perspective and propose the first gradient-based approach called DIFER (DIfferentiable automated Feature EngineeRing). We first propose an evolutionary framework to generate better features iteratively. Then, in each feature evolution step, we propose a tree-like structure called *parse tree* to represent constructed features

flexibly, and leverage a feature optimizer based on the encoder-predictor-decoder. Specifically, instead of searching in the discrete feature space, the encoder maps the traversal string of the parse tree into a continuous vector space. Constructing a better feature is equivalent to generating better embedding in the continuous vector space. The following predictor takes the feature embedding as input, predicts its performance score, and directly optimizes the embedding by gradient ascent along the score direction. The optimized embedding is further decoded as a better feature in the discrete space.

Extensive experimental results on both classification and regression tasks reveal that DIFER is not only effective but also efficient. Compared to the SOTA approach, DIFER achieves better performance on 22 out of 25 datasets with 40 times fewer feature evaluations. Moreover, DIFER can be effective when using different machine learning algorithms.

To summarize, our main contributions can be highlighted as follows:

- We propose a feature evolution framework to search for better features iteratively.

- To represent constructed features, we design the *parse tree* structure, which is more flexible and expressive than the commonly-used sequence representation.

- We introduce a novel feature optimizer based on the encoder-predictor-decoder for feature evolution and thus can achieve differentiable AutoFE. To our best knowledge, DIFER is the first differentiable AutoFE method.

- Extensive experimental results on a variety of tasks demonstrate that DIFER outperforms the state-of-the-art AutoFE approach in terms of both model performance and computational efficiency.

## 2 Related work

Feature engineering aims to transform raw data into features that can better express the nature of the problem. Recently, feature engineering has gradually shifted from leveraging human knowledge to automated methods. Existing AutoFE approaches can be divided into three categories.

**Heuristic Approaches**: Deep Feature Synthesis (DFS), the component of Data Science Machine (Kanter and Veeramachaneni, 2015), first enumerates all transformations on all features and then performs feature selection directly based on the improvement of model performance. One Button Machine (Lam et al., 2017) adopts a similar approach. However, this *expansion-reduction* approach suffers from a severe computational performance bottleneck due to the huge feature evaluation overhead. To avoid enumerating the entire feature space, Cognito (Khurana et al., 2016) introduces a tree-like exploration of feature space and presents handcrafted heuristics traversal strategies such as breadth-first search and depth-first search. AutoFeat (Horn et al., 2019) iteratively subsamples features using beam search. However, heuristic approaches cannot learn from past experiences and thus has a low search efficiency.

**Learning-Based Approaches**: To explore feature space efficiently, learning-based AutoFE methods have been proposed. LFE (Nargesian et al., 2017) trains an MLP and recommends the most likely useful transformation for each raw feature. However, it does not support transformation composition and works only for classification tasks. TransGraph (Khurana et al., 2018) trains a Q-learning agent to decide which transformation should be applied. Due to performing each transformation on all features, TransGraph suffers from feature explosion and low computational efficiency.

**NAS-Based Approaches**: Neural Architecture Search (Elsken et al., 2019) has aroused significant research interests in the field of AutoML (He et al., 2020). The reinforcement learning-based NAS method (Zoph and Le, 2017) views the structure of a neural network as a variable-length string. Then, it uses a recurrent network as the controller to generate such strings and trains the controller with policy gradient. This approach can be adopted into AutoFE. For instance, NFS (Chen et al., 2019), the current SOTA AutoFE method, utilizes several RNN-based controllers to generate transformation sequences for each raw feature. However, evaluating enormous sequences results

in substantial computational overhead. Most importantly, due to the side effects of reducing binary transformations to unary ones, NFS cannot generate complex features like $\frac{A+B}{C-D}$.

To improve the computational efficiency of NAS, differentiable methods have been proposed. DARTS (Liu et al., 2018) relaxes the categorical choice to a softmax over all possible operations, leading to a differentiable learning objective. NAO (Luo et al., 2018) maps the discrete architecture space to a continuous hidden space and optimizes existing architectures in the continuous space.

The differentiable NAS methods bring more inspiration to AutoFE. In this paper, we propose the first differentiable AutoFE method called DIFER, which can efficiently construct useful low-order and high-order features with much fewer feature evaluations.

## 3 Methodology

### 3.1 Problem Formulation

Let $D = \langle F, y \rangle$ be a dataset with a target vector $y$ and $n$ $d$-dimensional instances $F = \{f_1, \cdots, f_d\}$, where $f_i \in \mathcal{R}^n$ is the $i$-th raw feature. We denote the performance of the machine learning model $M$ that is learned from $D$ and measured by an evaluation metric $L$ (e.g., F1-score or mean squared error) as $L_M(F, y)$. Without loss of generality, the higher $L_M$ indicates better model performance.

Furthermore, we apply the composition of transformations $t \in \mathcal{R}^n \times \cdots \times \mathcal{R}^n \to \mathcal{R}^n$ to features for constructing new features. Let $o$ denote the arity of the transformation $t$, we construct a new feature $\hat{f} = t\left(\hat{f}_1, \cdots, \hat{f}_o\right)$, where $\hat{f}_j$ denotes the $j$-th input of $t$ to construct $\hat{f}$ for $j \in \{1, \cdots, o\}$. Given a set of transformations with different arities $T = \{t_1, \cdots, t_m\}$, we define the feature space $F^T$ as follows: $\forall \hat{f} \in F^T$, $\hat{f}$ satisfies any of the following conditions:

- $\hat{f} \in F$

- $\exists t \in T, \hat{f} = t\left(\hat{f}_1, \cdots, \hat{f}_o\right)$, where $\hat{f}_1, \cdots, \hat{f}_o \in F^T$

Formally, let $\alpha(\hat{f})$ denote the *order* of the feature $\hat{f} \in F^T$, $\alpha(\hat{f})$ can be defined as:

$$\alpha(\hat{f}) = \begin{cases} 1 + \max_j \alpha\left(\hat{f}_j\right) & \hat{f} = t(\hat{f}_1, \cdots, \hat{f}_o) \\ 0 & \hat{f} \in F \end{cases} \tag{1}$$

For example, we use the composition of the unary transformation *square* and the binary transformation *divide* to construct *BMI* (Body Mass Index), whose order is 2, by *divide* (*weight*, *square* (*height*)) with the raw features *weight* and *height*.

Therefore, the goal of AutoFE is to find the set of constructed features $F^*$ that can achieve the best performance:

$$F^* = \arg\max_{\hat{F}} L_M(F \cup \hat{F}, y), \text{ s.t. } \hat{F} \subset F^T \tag{2}$$

In practice, we limit the order of features and search in the feature space $F_k^T = \{\hat{f} \mid \hat{f} \in F^T \wedge \alpha(\hat{f}) \leq k\}$ since the size of the original space is infinite (i.e., $|F^T| = \aleph_0$). we explore $F_k^T$ and search for top features ranked by the performance metric $L_M\left(F \cup \{\hat{f}\}, y\right)$ as $F^*$. Moreover, similar to most existing AutoFE methods (e.g., NFS (Chen et al., 2019), (Nargesian et al., 2017), and TransGraph (Khurana et al., 2018)), we also append the constructed features to $F$ to maximize the modeling performance for a given algorithm.

### 3.2 Overview of DIFER

As shown in Figure 1, we propose an evolutionary framework to achieve AutoFE. The overall framework is divided into three phases: population initialization, feature evolution, and feature selection.

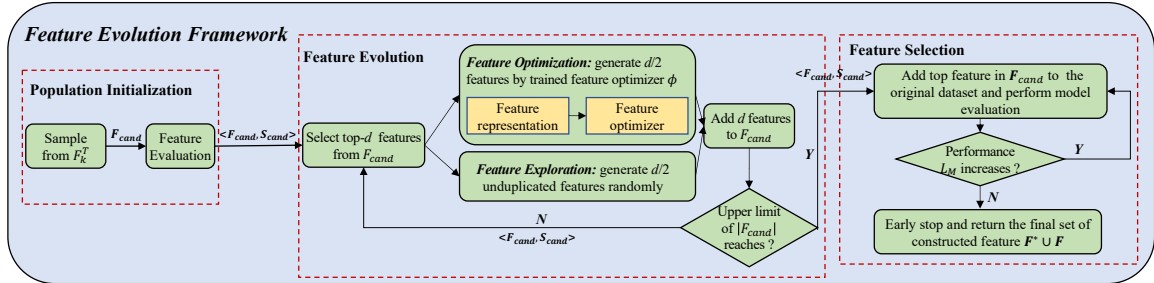

Figure 1: Overview of DIFER.

The population initialization phase constructs feature set $F_{\text{cand}}$ by randomly sampling features from $F_k^T$. We train a machine learner $M$, which takes instances as input and predicts the labels $y$, from scratch and evaluate its performance as the performance score of the feature $L_M(F \cup \{\hat{f}\}, y)$. Then, we can get the score set $S_{\text{cand}} = \{L_M(F \cup \{\hat{f}\}, y) | \hat{f} \in F_{\text{cand}}\}$.

The feature evolution phase aims to construct new features iteratively. In each iteration, we first select top-$d$ features from $F_{\text{cand}}$ according to $S_{\text{cand}}$. To enhance the diversity of evolution, we take two different approaches to generate new features at the same time. One way is to perform gradient-based optimization based on the feature optimizer and add $d/2$ optimized features to $F_{\text{cand}}$ (i.e., exploitation). The other way is to add $d/2$ unduplicated randomly-generated features to $F_{\text{cand}}$ for exploration. The process of feature evolution is repeated until a maximum number of feature evaluations is reached. In the feature optimization process, the two key components are the parse-tree-based feature representation and the gradient-directed feature optimizer that consists of an encoder, a predictor, and a decoder. Due to its flexibility in the optimization of complex feature transformation, the encoder-predictor-decoder-based feature optimizer is suitable for the AutoFE problem.

After the feature evolution phase, we select top features from $F_{\text{cand}}$ and add them to the original dataset. The number of added features is adaptively determined with an early-stopping mechanism. When the model performance no longer increases, we stop adding features to the original dataset.

**Case Study**. we show the process of DIFER using the dataset *PimaIndian* as an example. DIFER first initializes the population $\langle F_{cand}, S_{cand} \rangle$ by random sampling and evaluating features from $F_k^T$. Then, the feature optimizer is trained on the population. The detailed training process of feature optimizer is introduced in Section 3.4.

In the feature optimization process, taking the feature $\frac{min\_max(BloodPressure)}{Insulin}$ as an example, we introduce how the input feature is optimized to get a better feature. As mentioned in Section 3.3, the feature is first parsed as a tree and traversed to the string <Insulin,*Reciprocal*,BloodPressure,*MinMax,Multiply*>. The feature optimizer $\psi$ maps it into the continuous vector space as $e_x$ via the encoder $\psi_e$, optimizes the embedding $e_x$ along the gradient direction induced by the predictor $\psi_p$. The string <Insulin, Pregnancies, *AbsRoot, Multiply, Reciprocal*,BloodPressure,*MinMax,Multiply*> is recovered from the optimized embedding $e_{x'}$ by the decoder $\psi_d$. The recovered string is translated to $\frac{min\_max(BloodPressure)}{\sqrt{|Pregnancies| \cdot Insulin}}$.

### 3.3 Feature Representation

As shown in Figure 2, we design a tree-like structure called *parse tree* to represent constructed features. Compared with the sequence representation in NFS (Chen et al., 2019) and NAO (Luo et al., 2018), the parse tree is more flexible and expressive, which can represent complex *n*-ary feature transformation operation like $\frac{A+B}{C-D}$. The internal node in the parse tree indicates the transformation and the leaf node indicates the raw feature. We employ reversible post-order traversal to convert the

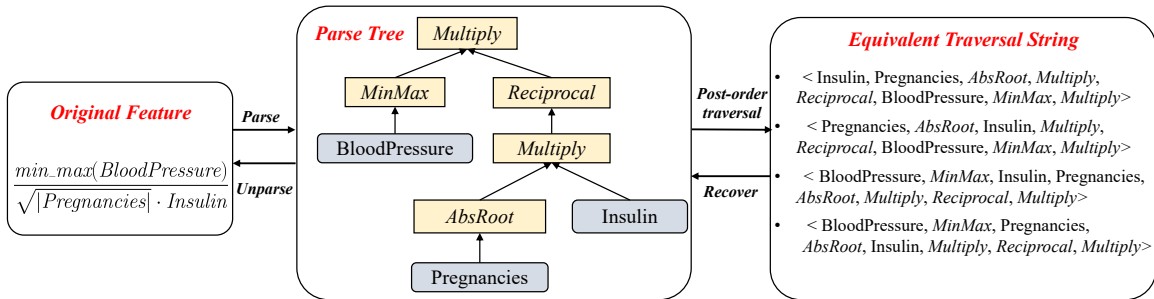

Figure 2: Parse tree and post-order traversal strings of the feature $\frac{min\_max(BloodPressure)}{\sqrt{|Pregnancies|} \cdot Insulin}$ in *PimaIndian*.

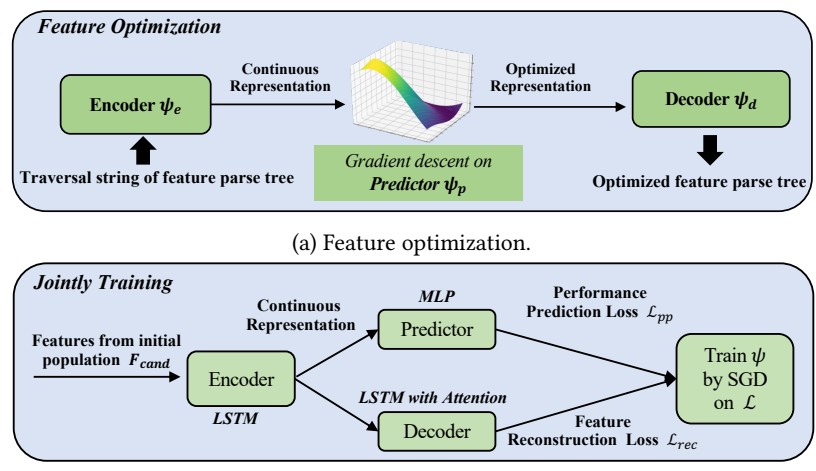

(a) Feature optimization.

(b) Jointly-training of feature optimizer.

Figure 3: Feature Optimizer of DIFER.

parse tree into equivalent traversal string $x$ as input to the encoder. The traversal string in Figure 2 shows an example where each word-based token (i.e., the original feature and the transformation) is separated by a comma. Let $x_r$ denote each token in the traversal string, where $r \in \{1 \cdots |x|\}$. Note that the relationship between the parse tree and the traversal string is one-to-many. When there are transformations where the input order is meaningless (e.g. $mul(a, b) == mul(b, a)$), the same parse tree can be converted into multiple equivalent strings. This nature can be viewed as a way of data augmentation when training the feature optimizer. Due to the fixed arity of each transformation, the optimized traversal string can be recovered to a parse tree with no ambiguity. The translation process can be found in Appendix A.

### 3.4 Feature Optimizer

DIFER employs a feature optimizer to construct new features based on the existing features. The feature optimization process is shown in Figure 3a. Specifically, the feature optimizer $\psi$ consists of an encoder $\psi_e$, a performance predictor $\psi_p$, and a decoder $\psi_d$. After jointly-training the feature optimizer for convergence, $\psi$ maps features into the continuous vector space via $\psi_e$, optimizes the embedding along the gradient direction induced by $\psi_p$, and recovers better features from the optimized embedding by $\psi_d$.

**Encoder**. The encoder $\psi_e$ maps the post-order traversal string $x \in \mathcal{X}$ to a continuous embedding $e_x \in \mathcal{E} \subset \mathcal{R}^{emb\_dim}$. Since the traversal string $x$ is a variable-length sequence, we use LSTM (Long Short-Term Memory) (Hochreiter and Schmidhuber, 1997) as the encoder. By the sum-

pooling technique, the sum of all hidden states $H_x = \{h_1, h_2, \cdots, h_{|x|}\}$ of the LSTM as the feature's continuous representation $e_x$.

**Predictor.** The predictor $\psi_p \in \mathcal{E} \to \mathcal{R}$ maps the continuous representation $e_x$ into its score $s_x$ measured by $L_M(F \cup \{\hat{f}_x\}, y)$. We employ a 5-layer fully-connected MLP as $\psi_p$.

**Decoder.** The decoder $\psi_d$ maps the embedding to the discrete feature space, i.e., the post-order traversal string of the optimized feature. According to the classical sequence-to-sequence method, we employ an LSTM with the attention mechanism (Bahdanau et al., 2015) as the decoder $\psi_d \in \mathcal{E} \to \mathcal{X}$, which takes $e_x$ as the initial hidden state and all hidden states $H_x$ in the encoder as the input of each timestamp.

**Jointly-Training.** To train the optimizer efficiently, we propose a jointly-training method based on a joint loss. The training dataset is the initial evaluated population $\langle F_{\text{cand}}, S_{\text{cand}} \rangle$. As shown in Figure 3b, we design a joint loss function that takes both the performance prediction loss $\mathcal{L}_{pp}$ and the structure reconstruction loss $\mathcal{L}_{rec}$ into account. The value of the hyperparameter $\lambda$ that balances $\mathcal{L}_{pp}$ and $\mathcal{L}_{rec}$ is determined adaptively (see Appendix C).

$$\mathcal{L} = \lambda \mathcal{L}_{pp} + \mathcal{L}_{rec}, \text{ where } \mathcal{L}_{pp} = \sum_x \left(s_x - \psi_p\left(\psi_e\left(x\right)\right)\right)^2 \text{ and } \mathcal{L}_{rec} = -\sum_x \sum_{r=1}^{|x|} \log P_{\psi_d}(x_r | \psi_e(x)) \quad (3)$$

### 3.5 Feature Optimization

After the convergence of the feature optimizer, we directly optimize the feature embedding $e_x$ in the continuous space by performing gradient ascent and then decode the optimized embedding into a new feature $x'$.

Starting from the constructed feature $x$, we optimize its embedding $e_x$ to get a better embedding along the gradient direction induced by the predictor $\psi_p$:

$$e_{x'} = \sum_{h_r \in H_x} \left(h_r + \eta \frac{\partial \psi_p}{\partial h_r}\right) \quad (4)$$

However, due to the nature that the corresponding parse tree of a feature $\hat{f}_x$ may have several equivalent post-order traversal strings $X = \{x^{(1)}, x^{(2)}, \cdots x^{(n)}\}$, the strings in $X$ are highly similar in the continuous space. After one step of gradient ascent, the decoded string of $e_{x'}$ may still be in $X$. Thus, we may get the same parse tree. We call $\eta$ in Equation (4) the evolution rate. Increasing the evolution rate $\eta$ can solve this problem to some extent (Luo et al., 2018). However, a large evolution rate would violate the preconditions of gradient ascent, resulting in no guarantee that $\psi_p(x + \Delta x) > \psi_p(x)$.

**Multi-step gradient ascent.** To address this problem, we propose a straightforward but effective strategy. Specifically, we apply the optimization process in Equation (4) multiple times with a small evolution rate $\eta$ until we get new parse trees. As a result, the number of times the optimization process (i.e., steps of gradient ascent) is adaptively determined. We refer to the overall process as feature optimization.

## 4 Experiments

### 4.1 Experimental Setting

As with the SOTA method NFS (Chen et al., 2019), we use 25 public datasets from OpenML (Vanschoren et al., 2014), UCI repository (Dua and Graff, 2017), and Kaggle (2021). There are 15 classification (C) datasets and 10 regression (R) datasets that have various numbers of features (5 to

10936) and instances (100 to 30000). In all experiments, we set the max order $k$ to 5 except in RQ3 and utilize 9 transformation functions totally. Moreover, to ensure the fairness, all methods except LFE (Nargesian et al., 2017) have the same feature transformation space.

- Unary transformation: *logarithm*, *square root*, *min-max normalization*, and *reciprocal*

- Binary transformation: *addition*, *subtraction*, *multiplication*, *division*, and *modulo*

All experiments are run using Tesla K80 (GPU) and Intel(R) Xeon(R) CPU E5-2630 v2 instances. To evaluate the AutoFE method, we use the performance metric $(1 − (relative\ absolute\ error))$ (Shcherbakov et al., 2013) for the regression task and *f1-score* for the classification task. 5-fold cross validation using random stratified sampling is employed and the average result is reported. Except that different ML algorithms are used in RQ4, we utilize Random Forest as default. We use scikit-learn as the machine learning algorithm library and employ PyTorch to implement the feature optimizer, including LSTM-based encoder and decoder, MLP-based predictor.

In the initialization step of DIFER, we randomly select 512 features as the initial population. Both the encoder and decoder of the feature optimizer are implemented as a one-layer LSTM. We empirically set the embedding size of each token in the traversal string and the size of the hidden state to 512. The predictor is a 5-layer MLP where the number of hidden units in each layer is 1024. To train the feature optimizer, we choose the Adam optimizer (Kingma and Ba, 2014) with a learning rate of 0.001 and a weight decay of 0.0001. The number of epochs is 400, and the batch size is 128. Early stopping is employed with a patience of 10.

In each feature evolution iteration, the value of $d$ is empirically set to be the minimum between top 20% of the initial population size and the total number of original features. The feature evolution runs until the number of feature evaluations reaches the upper limit of 4096. When optimizing the feature embedding in Equation (4), we perform gradient ascent with an evolution rate $\eta$ of 0.0001. Moreover, we use the same hyperparameters for all datasets. The robustness experiments with different hyperparameters can be found in Appendix D.

## 4.2 Effectiveness of DIFER (RQ1)

In this subsection, we demonstrate the effectiveness of DIFER. We compare DIFER on 25 datasets with the SOTA and baseline methods, including: (a) Raw: raw dataset without any transformation; (b) Random: randomly applying transformations to each raw feature; (c) DFS (Kanter and Veera-machaneni, 2015): a well-known *expansion-reduction* method; (d) AutoFeat (Horn et al., 2019): a popular Python library for automated feature engineering and selection; (e) LFE (Nargesian et al., 2017): recommend the most promising transformation for each feature using MLP; (f) NFS (Chen et al., 2019): the SOTA AutoFE method that achieves better performance than other existing approaches (e.g., Khurana et al. (2018)). The experimental settings of these methods, such as the set of transformations, the max feature order, and the evaluation metrics are the same as DIFER.

Table 1 shows the comparison results between DIFER and the existing methods. Moreover, since LEF can only deal with the classification task and the source code is not available, we directly use the best results reported in the original paper (Nargesian et al., 2017). The comparison results between DIFER and LEF are shown in Table 2. From Table 1 and Table 2, we can observe that:

- DIFER achieves the best performance in all but four cases. Although NFS greatly outperforms the baseline methods, DIFER still achieves an average improvement of 2.57% over NFS. For regression tasks, DIFER can even achieve a maximum improvement of 11.42%.

- DIFER can handle datasets with various numbers of instances and features for both regression and classification tasks and achieve performance improvement on all datasets with an average of 10.72% over Raw and an average of 9.55% over Random.

- With the benefit of searching in the continuous vector space, DIFER addresses the feature explosion problem while preserving the entire space, and achieves highly competitive performance even on large datasets such as *Credit Default* $(30000 \times 25)$ and *AP-omentum-ovary* $(275 \times 10936)$.

Table 1: Comparison between DIFER and the existing AutoFE methods (The datasets are sorted based on the evaluation time. † the results obtained using the open-sourced code, ∗ denotes statistically significant improvement measured by Friedman test and Nemenyi post-hoc test with $p$-value $< 0.05$. $\mathcal{T}$ indicates the total runtime. Inst. is short for Instance, Feat. is short for Feature, *Err.* indicates failure due to out of memory when running the open-source code).

| Dataset | C/R | Inst.\Feat. | Raw | Random | DFS† | AutoFeat† | NFS† | DIFER* | $\mathcal{T}_{NFS}$ | $\mathcal{T}_{DIFER}$ |
|---|---|---|---|---|---|---|---|---|---|---|
| Housing Boston | R | 506\13 | 0.4336 | 0.4446 | 0.3412 | 0.4688 | **0.5013** | 0.4944 | **566.42** | 982.15 |
| Bikeshare DC | R | 10886\11 | 0.8200 | 0.8436 | 0.8214 | 0.8498 | 0.9746 | **0.9813** | **595.57** | 1040.96 |
| Airfoil | R | 1503\5 | 0.4962 | 0.5733 | 0.4346 | 0.5955 | 0.6163 | **0.6242** | **603.80** | 1066.93 |
| Openml_586 | R | 1000\25 | 0.6617 | 0.6511 | 0.6501 | 0.7278 | 0.7401 | **0.7683** | 1722.49 | **1013.57** |
| Openml_589 | R | 1000\25 | 0.6484 | 0.6422 | 0.6356 | 0.6864 | 0.7141 | **0.7727** | 1726.04 | **1005.18** |
| Openml_637 | R | 1000\25 | 0.5136 | 0.5268 | 0.5191 | 0.5763 | 0.5693 | **0.6343** | 1411.79 | **1028.14** |
| Openml_618 | R | 1000\50 | 0.6267 | 0.6167 | 0.6343 | 0.6324 | 0.6400 | **0.6603** | 3159.47 | **1020.72** |
| Openml_607 | R | 1000\50 | 0.6344 | 0.6285 | 0.6388 | 0.6699 | 0.6870 | **0.6918** | 2990.91 | **1032.40** |
| Openml_616 | R | 500\ 50 | 0.5747 | 0.5714 | 0.5717 | 0.6027 | 0.5915 | **0.6554** | 1511.58 | **1030.57** |
| Openml_620 | R | 1000\25 | 0.6336 | 0.6178 | 0.6263 | 0.6874 | 0.6749 | **0.7442** | 1686.78 | **1047.37** |
| Hepatitis | C | 155\6 | 0.7860 | 0.8300 | 0.8258 | 0.7677 | 0.8774 | **0.8839** | **355.76** | 1045.77 |
| Fertility | C | 100\9 | 0.8530 | 0.8300 | 0.7500 | 0.7900 | 0.8700 | **0.9098** | **362.38** | 1054.51 |
| SpectF | C | 267\44 | 0.7750 | 0.8277 | 0.7906 | 0.8161 | 0.8501 | **0.8612** | **386.39** | 933.45 |
| Megawatt1 | C | 253\37 | 0.8890 | 0.8973 | 0.8773 | 0.8893 | 0.9130 | **0.9171** | **404.33** | 1024.95 |
| Ionosphere | C | 351\34 | 0.9233 | 0.9344 | 0.9175 | 0.9117 | 0.9516 | **0.9770** | **421.50** | 1036.01 |
| German Credit | C | 1001\24 | 0.7410 | 0.7550 | 0.7490 | 0.7600 | **0.7818** | 0.7770 | **433.39** | 1043.06 |
| Credit-a | C | 690\6 | 0.8377 | 0.8449 | 0.8188 | 0.8391 | 0.8652 | **0.8826** | **435.14** | 992.91 |
| PimaIndian | C | 768\8 | 0.7566 | 0.7566 | 0.7501 | 0.7631 | 0.7839 | **0.7865** | **435.10** | 1007.30 |
| Messidor_features | C | 1150\19 | 0.6584 | 0.6878 | 0.6724 | 0.7359 | 0.7461 | **0.7576** | **555.62** | 1069.04 |
| Wine Quality Red | C | 999\12 | 0.5317 | 0.5641 | 0.5478 | 0.5241 | **0.5841** | 0.5824 | **587.77** | 1033.29 |
| Wine Quality White | C | 4900\12 | 0.4941 | 0.4930 | 0.4882 | 0.5023 | 0.5150 | **0.5155** | 1278.61 | **1016.35** |
| SpamBase | C | 4601\57 | 0.9102 | 0.9237 | 0.9102 | 0.9237 | 0.9296 | **0.9339** | 993.92 | **959.03** |
| AP-omentum-ovary | C | 275\10936 | 0.7636 | 0.7100 | 0.7250 | *Err.* | 0.8640 | **0.8726** | 4183.75 | **1441.01** |
| Credit Default | C | 30000\25 | 0.8037 | 0.8060 | 0.8059 | 0.8060 | 0.8049 | **0.8096** | 9253.70 | **1204.99** |
| gisette | C | 2100\5000 | 0.9261 | 0.8710 | 0.7410 | *Err.* | 0.9590 | **0.9635** | 18877.07 | **1646.19** |
| Upper Limit of Eval. Num. | | | | | | | 160,000 | **4,096** | | |

**Effectiveness of the predictor** $\psi_p$. Since the accuracy of the predictor determines the quality of the optimized features, here we demonstrate the effectiveness of $\psi_p$. We train the feature optimizer using the data augmentation technique mentioned in Section 3.3 on an initialized population of 512 features. After convergence, the loss $\mathcal{L}_{pp}$ (i.e., Mean-Squared Error) of the predictor in the training set is 0.00106. To test the predictor, we randomly sample 256 features from the feature space as the test set, which is different from the training set. The test loss of $\psi_p$ is 0.00132. Both the training loss and the test loss are small and close, demonstrating the effectiveness of the predictor. Furthermore, we employ the pairwise accuracy metric to evaluate $\psi_p$. Let $X$ denote the test set. $f(x)$ and $y$ denote the predicted performance of $\psi_p$ and the real performance of the feature. The pairwise accuracy is defined as follows:

$$pairwise\ accuracy = \frac{\sum_{x_1 \in X, x_2 \in X} \mathbb{I}_{f(x_1) \geq f(x_2)} \mathbb{I}_{y_1 \geq y_2}}{|X|(|X| - 1)/2} \tag{5}$$

where $\mathbb{I}$ represents the 0-1 indicator function. The pairwise accuracy of $\psi_p$ is 0.918, which is close to the ideal value (i.e., 1) and much better than random guess (i.e., 0.5).

## 4.3 Efficiency of DIFER (RQ2)

The overhead of AutoFE can be divided into two parts: the process of feature evaluation and the training overhead of the controller (i.e., the feature optimizer). To verify the efficiency of DIFER, we conduct experiments in terms of the total runtime and the number of feature evaluations, respectively. Table 1, where the datasets are sorted in ascending order of model evaluation time, shows the total runtime $\mathcal{T}$ and the average number of feature evaluations for AutoFE, and Figure

| Dataset | LFE* | NFS† | DIFER |
|---|---|---|---|
| Credit-a | 0.771 | 0.8652 | **0.8826** |
| Feritility | 0.873 | 0.8700 | **0.9098** |
| Hepatitis | 0.831 | 0.8774 | **0.8839** |
| Ionosphere | 0.932 | 0.95160 | **0.9770** |
| Megawatt1 | 0.894 | 0.9130 | **0.9171** |
| SpamBase | **0.947** | 0.9296 | 0.9339 |

Table 2: Comparison between DIFER, LFE, and NFS (* the results reported in the paper).

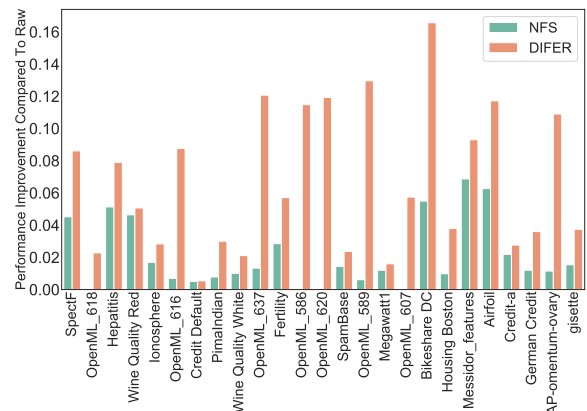

Figure 4: Comparison between NFS and DIFER. The number of feature evaluations is restricted to 3500.

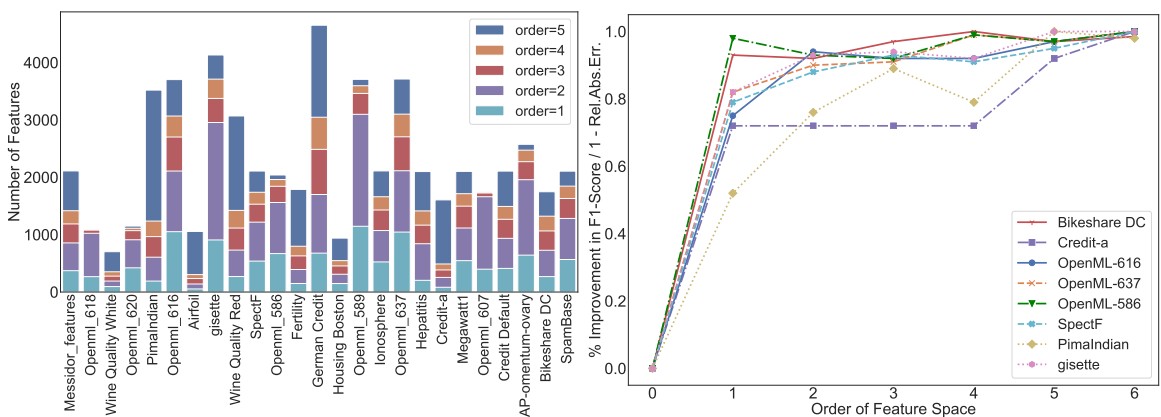

(a) Proportion of different order features generated by DIFER.

(b) Effect of the high-order feature space.

Figure 5: Effectiveness of high-order features.

4 shows the comparison results between NFS and DIFER with a restricted number of feature evaluations. From Table 1 and Figure 4, we can observe that:

- In Table 1, DIFER achieves better performance than NFS by using 40 times fewer feature evaluations while still achieving significant performance improvement.

- From the perspective of runtime, the overhead of DIFER is mainly in the training and inference of the feature optimizer compared to NFS which is dominated by feature evaluation. Therefore, the efficiency advantage of DIFER is more obvious on larger datasets that requires more evaluation time. For example, compared with NFS, DIFER can achieve 2.9×, 7.7×, 11.5× speedup on *AP-omentum-ovary*, *Credit Default*, *gisette*, respectively.

- The advantage of DIFER is more significant with a restricted number of feature evaluations measured by Wilcoxon signed-rank test with $p$-value $< 0.05$. DIFER achieves an average performance improvement of 6.89%, doubling that in RQ1.

### 4.4 Effectiveness of High-Order Features (RQ3)

To evaluate the ability of exploring the high-order feature space, we conduct two experiments:

1. Analyze the features generated by DIFER during the search process and investigate whether DIFER can indeed search for the high-order features.

Table 3: Statistics on the performance of DIFER with different ML algorithms.

| Task | Algorithm | Avg Impr±Std (%) | Min/Max Impr (%) |
|---|---|---|---|
| Classification | RandomForest | 6.59±4.23 | 0.73 / 15.06 |
| | LogisticRegression (Hosmer Jr et al., 2013) | 5.95±4.12 | 1.01 / 15.94 |
| | LinearSVC (Cortes and Vapnik, 1995) | 13.98±9.23 | 3.17 / 22.32 |
| | XGBoost (Chen and Guestrin, 2016) | 6.90±6.92 | 0.30 / 27.98 |
| | LightGBM (Ke et al., 2017) | 7.69±8.09 | 0.16 / 32.63 |
| Regression | RandomForest | 16.42±6.19 | 5.36 / 25.80 |
| | LassoRegression (Tibshirani, 1996) | 14.61±8.92 | 1.22 / 66.66 |
| | LinearSVR (Smola and Schölkopf, 2004) | 32.72±19.79 | 13.21 / 96.98 |
| | XGBoost (Chen and Guestrin, 2016) | 13.47±9.35 | 3.20 / 67.06 |
| | LightGBM (Ke et al., 2017) | 15.46±10.48 | 4.75 / 71.92 |

2. Choose the max order $k$ from 0 to 6, where $k = 0$ means the raw dataset without any feature transformation. Then, we analyze the performance curve by varying $k$.

Figure 5a shows the number of each order features generated by DIFER with $k = 5$ for each dataset. High-order features take a considerable average proportion of 80.9%, confirming that DIFER exploits the entire feature space $F_k^T$ instead of its subspace $F_i^T$ where $i < k$.

Besides, we randomly choose 8 datasets, normalize the performance of DIFER, plot the performance curve with the increasing max order in Figure 5b, and draw the following conclusions:

- The overall performance of DIFER stably increases with the max order $k$. However, when $k$ increases to 5, performance improvement become insignificant.

- For most datasets, sufficient performance improvement can be already achieved with $k = 2$. There is no need to set an excessively large max order in practice.

### 4.5 Different Machine Learning Algorithms (RQ4)

To further investigate the generalization ability of DIFER, we utilize the commonly-used classification and regression algorithms. We conduct experiments on all datasets and the performance statistics are shown in Table 3. Compared to the Raw method, DIFER achieves significant improvement under different algorithms. For instance, LinearSVR with DIFER even achieves an average improvement of 32.72% across 25 datasets and a max improvement of 96.98% in *Airfoil*.

## 5 Conclusion and Future Work

In this work, we proposed DIFER, to the best of our knowledge, the first differentiable AutoFE method. DIFER leverages an encoder-predictor-decoder-based feature optimizer, which maps features into the continuous vector space via the encoder, optimizes the embedding along the gradient direction induced by the predictor, and recovers better features from the optimized embedding by the decoder. Moreover, based on the feature optimizer, we proposed a feature evolution framework to search for better features iteratively. Experimental results show that DIFER is effective on both classification and regression tasks and can outperform the existing AutoFE methods in terms of both prediction performance and computational efficiency.

The transformation operations in DIFER are for numerical features. For future work, we plan to automatically search for transformations for different feature types (i.e., numerical and categorical).

**Acknowledgements**. This work was supported in part by the National Natural Science Foundation of China (#62102177 and #U1811461), the Natural Science Foundation of Jiangsu Province (#BK20210181), the Key Research and Development Program of Jiangsu Province (#BE2021729), and the Collaborative Innovation Center of Novel Software Technology and Industrialization, Jiangsu, China.

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

## A  Translation Between Three Forms of Features

As mentioned in Section 3.3, there are three forms of features (i.e., original form, parse tree form and traversal string form). To generate parse trees from the original features, we design the following context-free grammar in BNF (Backus Normal Form):

- $ParseTree := f_{1,\dots,d} \mid UnaryOp\ ParseTree \mid BinaryOp\ ParseTree\ ParseTree$

- $UnaryOp := logarithm \mid abs \mid root \mid min\text{-}max \mid normalization \mid reciprocal$

- $BinaryOp := addition \mid subtraction \mid multiplication \mid division \mid modulo$

Through such syntax parser, the features are parsed into the form of parse tree, and then the corresponding strings is derived through post-order traversal. Similarly, due to the many-to-one relationship between traversing strings and parse trees, strings can be reduced to parse trees. With features as leaf nodes, the constructed features are finally obtained from the bottom up at the root node through the internal nodes with the operators.

## B  Neighborhood of Constructed Features

The duplicated post-order traversal strings $X = \{x^{(1)}, x^{(2)}, \cdots x^{(n)}\}$ of the feature $\hat{f}_x$ are highly similar in the continuous space:

$$\exists \epsilon,\ \forall x^{(i)} \in X, \|e_{x^{(i)}} - \frac{1}{n} \sum_{x^{(j)} \in X} e_{x^{(j)}}\|_2 \le \epsilon \tag{6}$$

where $\epsilon$ is just a variable used to inscribe the property, not a hyperparameter. The neighborhood of $X$ in the continuous space can be represented as $\delta_X = \{e_{x'} \mid \|e_{x'} - \frac{1}{n} \sum_{x^{(j)} \in X} e_{x^{(j)}}\|_2 \le \epsilon\}$. After one step of gradient ascent, the decoded string of $e_{x'} \in \delta_x$ may still be in $X$. Thus, the same parse tree is got. To escape the neighborhood., we use multi-step gradient ascent as mentioned in Section 3.5.

## C  Adaptive Loss Weight Setting

We use the parameter $\lambda \in \mathcal{R}^+$ to balance $\mathcal{L}_{pp}$ and $\mathcal{L}_{rec}$ and $\lambda$ is determined adaptively. Inspired by (Goyal et al., 2017), the first $k$ epochs are used to warm up the jointly-training of the feature optimizer with $\lambda = 1$. After the first $k$ epochs, we assign $\lambda = \sum_{i=1}^{k} \mathcal{L}_{rec} / \sum_{i=1}^{k} \mathcal{L}_{pp}$ according to the sum of losses. This is mainly to make the two losses in the same order of magnitude. In practice, $k$ is empirically set to 5.

## D  Robustness

We further evaluate whether DIFER is sensitive to different hyperparameters, including evolution rate $\eta$ and population size $p$. Figure 6a and Figure 6b show the experimental results on 5 randomly-selected datasets that represent both classification and regression tasks. Figure 6a demonstrates that DIFER is robust to different settings of $\eta$. Empirically, $\eta$ should not be too large in gradient ascent. A small $\eta$ can get the same or even better results than the large $\eta$ by performing multiple times of gradient ascent. Moreover, a larger $p$ allows the feature optimizer to be fully trained, and a smaller $p$ allows more features to be optimized in the case of a limited number of feature evaluations. As shown in Figure 6b, the performance of DIFER remains stable across different settings of $p$.

## E  Statistics Comparison

To further statistically evaluate the difference between the AutoFE methods in Table 1, we perform the Friedman test Demšar (2006), which is a non-parametric equivalent of the repeated-measures ANOVA. It is used to determine whether or not there is a statistically significant difference.

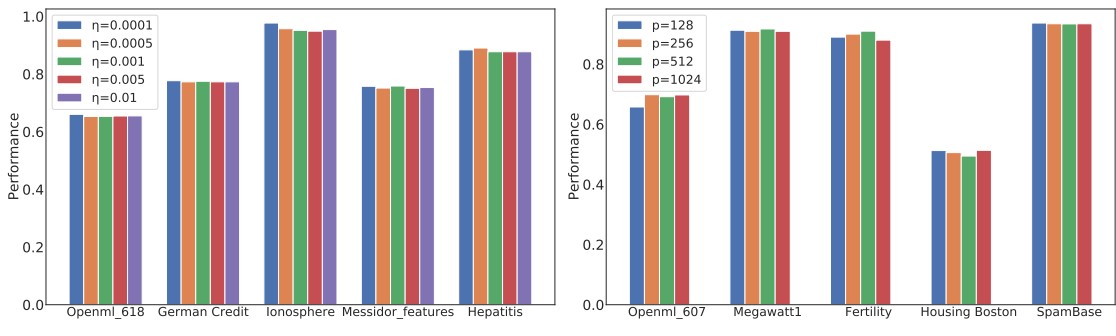

(a) Comparison results of DIFER with different settings of the evolution rate $\eta$.

(b) Comparison results of DIFER with different settings of the population size $p$.

Figure 6: Robustness of DIFER.

Table 4: $p$-values for each pairwise comparison using the Nemenyi post-hoc test for the AutoFE methods (Confidence level $p = 0.05$).

|          | DFS    | AutoFeat | NFS    | DIFER  |
|----------|--------|----------|--------|--------|
| DFS      | 1.0000 | 0.2967   | 0.0010 | **0.0010** |
| AutoFeat | 0.2967 | 1.0000   | 0.0313 | **0.0010** |
| NFS      | 0.0010 | 0.0313   | 1.0000 | **0.0046** |
| DIFER    | **0.0010** | **0.0010** | **0.0046** | 1.0000 |

For the comparison results in Table 1, we first calculate the Friedman statistic. Let $r_i^j$ be the rank of the $j$-th of $k$ AutoFE methods ($k = 4$, i.e., DFS, AutoFeat, NFS, and DIFER) on the $i$-th of $N$ datasets. The Friedman test compares the average ranks of models, $R_j = \frac{1}{N} \sum_i r_i^j$. The null-hypothesis states that all the tree models are equivalent and so their ranks $R_j$ should be equal. We employ the scipy tool[1] to calculate the Friedman statistic. The corresponding Friedman $p$-value is 1.17e-10. Since the $p$-value is less than 0.05, we can reject the null hypothesis that the performance is the same for all four types of AutoFE methods. In other words, we have sufficient evidence to conclude that the AutoFE method lead to statistically significant differences in terms of performance. Since the $p$-value of the Friedman test is statistically significant, we perform the Nemenyi post-hoc test Nemenyi (1963) to further determine exactly which AutoFE method has different means. Table 4 shows the $p$-values for each pairwise comparison. We can conclude that DIFER is significantly different from other trees for a confidence level of $p = 0.05$ and show the result by '*' in Table1.

## F Numbers of Added Features

In the formal definition, $|F^*|$ is defined as the set of added features, which does not contain the raw features. Table 5 shows the number of features finally added by AutoFeat, NFS and DIFER. For NFS, it is the number of original features. Benefiting from feature selection, $|F^*|$ in AutoFeat and DIFER is adaptive.

---

[1]https://github.com/scipy/scipy

Table 5: The number of features $|F^*|$ added into the original dataset by AutoFE methods (*Err.* indicates failure due to out of memory when running the open-source code).

| Dataset | C/R | Inst.\Feat. | $|F^*|_{AutoFeat}$ | $|F^*|_{NFS}$ | $|F^*|_{DIFER}$ |
|---|---|---|---|---|---|
| Housing Boston | R | 506\13 | 21 | 13 | 1 |
| Bikeshare DC | R | 10886\11 | 17 | 11 | 6 |
| Airfoil | R | 1503\5 | 4 | 5 | 4 |
| Openml_586 | R | 1000\25 | 37 | 25 | 20 |
| Openml_589 | R | 1000\25 | 21 | 25 | 20 |
| Openml_637 | R | 1000\25 | 30 | 25 | 13 |
| Openml_618 | R | 1000\50 | 49 | 50 | 32 |
| Openml_607 | R | 1000\50 | 51 | 50 | 38 |
| Openml_616 | R | 1000\50 | 41 | 50 | 8 |
| Openml_620 | R | 1000\50 | 32 | 50 | 12 |
| Hepatitis | C | 155\6 | 7 | 6 | 6 |
| Fertility | C | 100\9 | 12 | 9 | 3 |
| SpectF | C | 267\44 | 37 | 44 | 9 |
| Megawatt1 | C | 253\37 | 48 | 37 | 29 |
| Ionosphere | C | 351\34 | 52 | 34 | 1 |
| German Credit | C | 1001\24 | 22 | 24 | 1 |
| Credit-a | C | 690\6 | 4 | 6 | 5 |
| PimaIndian | C | 768\8 | 12 | 8 | 1 |
| Messidor_features | C | 1150\19 | 29 | 19 | 10 |
| Wine Quality Red | C | 999\12 | 8 | 12 | 6 |
| Wine Quality White | C | 4900\12 | 11 | 12 | 9 |
| SpamBase | C | 4601\57 | 46 | 57 | 1 |
| AP-omentum-ovary | C | 275\10936 | *Err.* | 10936 | 491 |
| Credit Default | C | 30000\25 | 30 | 25 | 5 |
| gisette | C | 2100\5000 | *Err.* | 5000 | 19 |

