# OpenReview forum: "DIFER: Differentiable Automated Feature Engineering"
_automl.cc/AutoML/2022/Track/Main — AutoML-Conf 2022 (Main Track)_

### Official Review · Reviewer_ibk3 · 2022-04-01

**Potential Impact On The Field Of Automl Rating:** 4
**Technical Quality And Correctness Rating:** 2
**Clarity Rating:** 3

**Summary Of Contributions:**

This paper introduces a novel approach to automated feature engineering. In order to tackle the computational complexity of finding the appropriate sequence of data transformations, they apply the method of mapping the discrete decision space to a continuous space. This allows to scale the search for which features to add. They show how their approach performs in various experimental evaluations.

**Clarity:**

See technical clarity.

Also better not to use expressions such as 'brutally' - significantly could probably work.
'no longer increases with a specific patience' = maybe 'specific limit on the number of iterations'?

**Overall Review:**

The approach employed here is based on the technique employed in Neural Architecture Optimize (NAO) as cited by the authors [12]. DIFER encodes the features into a continuous space. NAO encodes the neural network architecture into a continuous space. Both perform gradient descent in the direction to increase the predictors accuracy and the decoder is used for stability so that the encoder is embedding the features correctly. Then the loss function is also the same for both DIFER and NAO: reconstruction loss + predictors RMSE loss. There are some differences as the authors pointed out in some other rebuttal, and I think applying a technique in a different context is somewhat ok.

And credit to the authors, they did address quite a few issues over time in this paper (such as adding the number of added features in the table; trying different models). However, one other point that was made by various reviewers was the runtime complexity. It is fine to show the number of iterations, but they are not as revealing as the actual run time for the proposed method. Would it be possible to just add another column on runtime? I do not expect the numbers to be much worse than the ones obtained by other approaches such as RL based methods. Of course, an approach like LFE has a clear runtime advantage but also disadvantages.
Overall I think it would be really useful to expose the computational complexity of the method.

To me there is also still a bit of a lack of understanding of encoder/decoder and how all the pieces with together. Yes, I think I do understand it in general - maybe you need a concrete example that shows what exactly happens at each major component such as the embedding space and how to encode and decode? Maybe one issue is that there are so many parameters on how to create the parse tree, extend it etc, that it just becomes difficult to have a clear picture. And maybe it is just mostly performance driven anyway and that is just how things are selected in the end.
Do you also know how accurate the predictions by the MLP actually are?

One other generic question I had is on why those rather simple transforms cannot be learned by an estimator? For instance, it is always claimed that Neural Networks learn their own feature representation - but they cannot learn a product and sum transform? Anyway, that is not really a question that is important for the acceptance of this paper. This brings of course the idea to mind to based on your work create an end to end NN that selects transforms but also makes the actual predictions at the end.


**Potential Impact On The Field Of Automl:**

Automated feature engineering is very important for practical relevance of AutoML.
The approach here is in general reasonable and is inspired by a NAS based-approach which gets quite some citations.

**Reproducibility:**

Seems mostly reproducible given that now the code is available too.

**Review Confidence:**

4: You are confident in your assessment, but not absolutely certain. It is unlikely, but not impossible, that you did not understand some parts of the submission or that you are unfamiliar with some pieces of related work.

**Review Rating:**

3: Marginally below the acceptance threshold (use sparsely)

**Review Summary:**

The paper does adopt a method from NAS for AutoFE and is over all interesting.
I do still lack a bit of clarity on the approach including its computational complexity.

**Technical Quality And Correctness:**

While the main approach seems somewhat clear to me, there are some details that I am still missing.
What are the criteria to pick features by the evolutionary algorithm?
How costly is the training of the predictor? What is the computational complexity?
A column that shows runtime would certainly help.

---

### Official Review · Reviewer_XVwV · 2022-04-04

**Potential Impact On The Field Of Automl Rating:** 3
**Technical Quality And Correctness Rating:** 3
**Clarity Rating:** 4

**Summary Of Contributions:**

The authors propose an algorithm for automated differentiable feature engineering.
The algorithms utilize a basic evolutionary framework hybridized with and encoder-predictor-decoder neural network for feature optimization, which is trained via gradient descent, making the search space effectively differentiable.

**Clarity:**

The paper is clearly structure and well written.
Concepts are explained compactly but in sufficient detail.

**Ethics Details (Optional):**

Unfortunately, no direct discussions on the ethics of feature engineering was provided.
I am aware of the fact that feature engineering itself is an abstract task and so is gradient descent.
However, previous works have demonstrated [1] that gradient descent-based systems, depending on the data they are trained with
may produce unwanted biases. It would be interesting to elaborate on how and whether such biases could be inductively induced into a model by trained feature engineering.


References
Burns, Kaylee et al. “Women also Snowboard: Overcoming Bias in Captioning Models.” ArXiv abs/1803.09797 (2018): n. pag.



**Overall Review:**

The paper is well written and structured with a well-maintained code repository allowing for easy reproducibility.
The authors propose an innovative, state-of-art-performance achieving method for automated feature engineering.
However, the evaluation regarding the efficiency of the model is lacking, and ethics statement is missing in any capacity.


**Potential Impact On The Field Of Automl:**

Feature Engineering is, besides data cleaning, one of the most time-consuming tasks in general machine learning that is also heavily reliant on expert knowledge.
Providing an efficient and automated solution to the problem could potentially be highly beneficial to data scientists and the field of AutoML in general. Differentiable Neural Architecture Search has shown that gradient descent can be leveraged effectively to solve problems within the general domain of AutoML surprisingly efficient. For this reason, I think an attempt to create a differentiable automatic feature engineering algorithm is a sound innovation in automated feature engineering that is citation worthy.

**Reproducibility:**

A set of bash scripts is provided to reproduce the experimental results on individual datasets and models.
The code is well-structured and nicely written.
Requirements.txt and instructions are provided.



**Review Confidence:**

3: You are fairly confident in your assessment. It is possible that you did not understand some parts of the submission or that you are unfamiliar with some pieces of related work.

**Review Rating:**

5: Accept, good paper

**Review Summary:**

Overall, I really enjoyed reading the paper. It provides a novel and interesting approach to a relevant problem while providing state-of-the-art performance.
The remaining concerns I raised during this review can be addressed easily in the next phase of the review.

**Technical Quality And Correctness:**

The paper is overall well written and structured. The algorithm is explained in detail and the illustrations aid in the understanding of the hybridization between an evolutionary-like loop and the differentiable parts which involves and lstm-encoder and decoder as well as an MLP also operating on the encoders output for performance prediction.
The experiments orient themselves on the methodology of previous state-of-the-art proposals and provide extensive comparisons with other algorithms on numerous datasets.
The results demonstrate a consistent improvement over the state of the art in predictive quality.

However, since the expensiveness and computational complexity of previous algorithms was mentioned, it would have been nice to also include the computation time for each of these benchmarks or some additional studies on efficiency. In other words, I find section 4.3 lacking, since I am uncertain whether this increased efficiency is reflected in less computational resources for the generating the SOTA-results in Table 1.

---

### Official Review · Reviewer_8xBx · 2022-04-04

**Potential Impact On The Field Of Automl Rating:** 3
**Technical Quality And Correctness Rating:** 3
**Clarity Rating:** 3

**Summary Of Contributions:**

The authors present a differentiable representation that improves evolutionary search for feature transformations.  The method combines a number of learning techniques into a single algorithm for automatic feature transformation learning.  Empirical comparison with other AutoFE methods suggest that their method (DIFER) yields more accurate feature transformations on a variety of benchmark datasets and is relatively efficient.

**Clarity:**

The paper is not poorly written, but I had difficulty understanding in detail, and also intuitively, exactly how and why the method works.  It looks like most of the details are there, but little effort has been made to explain how these methods work together and why the specific choices of methods were made.

**Overall Review:**

The method proposed in this paper for AutoFE is interesting.  But it is also complex because it combines a number of learning algorithms in a complex system, and it is not clear what components of this system are critical for good performance, and which are not.  Overall it is difficult to understand exactly how and why the method works.  It would be good if the paper included concrete examples of the result of feature engineering and how different components of the system led it to devise those features.  The empirical results look reasonably strong, though there are some questions about the way computational performance is compared across competing methods.

The positive aspects of the paper are the use of a differentiable representation for AutoFE, and performing a fairly comprehensive empirical comparison.  the negative aspects of the paper are the clarity with which the method is explained and justified, and the lack of concrete examples that would further motivate the algorithm and the results of AutoFE.

**Potential Impact On The Field Of Automl:**

AutoFE is an important subdiscipline within AutoML, and improved methods in AutoFE would be an important contribution to improving AutoML.  The results in this paper suggest that by moving to a continuous and differentiable representation of the feature engineering space, that significant improvements in AutoFE can be gained, analagous to improvements that have been made in NAS using continuous and differentiable representations of neural architectures.

**Reproducibility:**

I'm not sure the level of detail presented in the paper would make it easy for other researchers to reproduce the method and results.  But their code is available online and should allow the results to be mechanically reproduced.

**Review Confidence:**

2: You are willing to defend your assessment, but it is quite likely that you did not understand the central parts of the submission or that you are unfamiliar with some pieces of related work.

**Review Rating:**

3: Marginally below the acceptance threshold (use sparsely)

**Review Summary:**

An interesting AutoFE method based on a differentiable representation that appears to outperform existing AutoFE methods.  Although it is not clear exactly how the components of this algorithm work together to generate those results, the results look good.  Perhaps if the paper did a better job explaining why and how the algorithm works, it would be easier to recommend it.

**Technical Quality And Correctness:**

"More data beats clever algorithms, but better data beats more data. Feature engineering, the process of constructing features from raw data, directly determines the upper bound of machine learning algorithms. However, it requires considerable domain knowledge to construct features."

Not sure everyone would agree with that.  Some users of DNNs/CNNs/... believe the trick is to give large data with raw features to the deep models and allow them to automatically learn appropriate internal feature representations.  I'm not arguing against your work, but this is a provocative way to start the paper that not everyone would agree with.

When comparing runtime cost, perhaps you should use wall clock time to make the comparison more fair?  Leaving out part of the computation needed for DIFER seems questionable.

If you are going to use learning methods such as XGB and LightGBM, why include feature transformations such as log which by themselves can't help decision tree algorithms?  Is it because log transforms can be combined with other transforms such as addition and multiplication?

---

### Official Review · Reviewer_h3zi · 2022-04-05

**Potential Impact On The Field Of Automl Rating:** 3
**Technical Quality And Correctness Rating:** 3
**Clarity Rating:** 2

**Summary Of Contributions:**

The authors propose a differentiable AutoFE method, called DIFER, that is capable to produce high-order features. The general idea is to evolve a set of features, starting with the original features of the given dataset, so that in each iteration new features are added, either at random or based on the best performing features currently in the set. In the latter case, a so called "gradient-directed feature optimizer" is used, which consists of 1) an encoder that transforms a traversal string of the parse tree representation of a feature into a continuous representation (dense numerical representation of the string); 2) a predictor that estimates the performance of the continuous representation of the feature; 3) a decoder that transforms the continuous representation back into a traversal string of a parse tree. The main idea of this feature optimizer is to adapt the continuous representation of a feature, based on a gradient, until the predictor estimates a way better performance. Then the decoder is used to transform the continuous representation back into the parse tree of that feature. Note that the predictor is trained once in the beginning. This process is repeated until some maximum number of iterations is reached. The finally returned set of features is constructed based on the original features. Additionally, in each iteration the best performing feature from the previously grown set of features is selected for the final set, as long as there is improvement when adding the best performing feature.

The authors provide experiments of DIFER compared to 6 competitors, containing the effectiveness of DIFER, the efficiency of DIFER in terms of pipeline evaluations, the effectiveness of features having different (higher) order, as well as experiments covering different machine learning algorithms.

**Clarity:**

Although the presented idea seems to be good, the way it is presented has a lot of improvement potential (especially Sec. 3).

First, the formal definition is not well written. Some formulas or equations are given, and seem to be correct, but I am missing contextual explanations of the formulas, i.e. what do they mean? The given example is great for understanding the general problem, but I am missing the relation to the variables for a complete example.

Second, the overview of DIFER is missing some information, like the naming of the different phases, which are referred to later. Here all components should be named and their interaction should be made very clear, so that in the following subsections just the components are described. Additionally, the Fig. 1a has a lot of improvement potential, starting with the contrary coloring (see box "Feature Evaluation" in grey and the green color explanation "Feature evolution step"), the missing stopping criterion, as well as the missing creation of the finally returned set of features. In general, the variable usage is not consistent in this image, as e.g. S_cand is missing. Furthermore, the usage of text in box and next to arrows should be used in a consistent way, as the Feature optimization/exploration should also be written in box according to its action kind. On the other hand, the information about the figure 1b is not well placed here, it belongs into the according section.

Third, the feature representation section should make clear how the tokens are determined (e.g. on a letter or word level). I am actually wondering, if an order can be defined either on the parse tree construction or on the traversal string construction to prevent the 1:n relation. On the other hand, with the given argumentation I am wondering why there is no m:n relation.

Fourth, the description of the feature optimizer is lacking a lot of contextual information. In that subsection, the actual idea on an abstract level should be explained. i.e. what is the goal (learning a score predictor based on a numerical representation of the feature, and having according encoder/decoder). Maybe rethinking about the naming of the "feature optimizer" could help to clarify what this component is meant to do, e.g. feature score/performance predictor. The "how is it used" is then what should be explained in an abstract way at the beginning of the optimization section.

In general, the order of the figures does not match their occurrence in the text, the alignment seems to be wrong (Fig. 2 not top), Table 3a is actually a figure and therefore also the order and numbering of the tables does not fit. Figure 3b is too small to read the text without scrolling, both Fig. 3b and 4a do not contain all datasets.

The number of evaluations is a good foundation, but the description of results / insights is often very short.

**Overall Review:**

Overall, the idea of the presented paper is very interesting. Unfortunately, the descriptions are often very short and for someone who did not see such a usage of an encoder decoder before, hard to understand. The images are missing important information, the ordering of the figures and tables is not reasonable. The evaluation is a good foundation, but the some experiment setups do not provide a fair comparison or do not cover all aspects or datasets. All in all, this paper seems to be not carefully finalized.

The final decision will be dependent on the improvements on the methodology and evaluation (Sec. 3 and Sec. 4). If the concerns are tackled, I am willing to fully accept the paper.

**Potential Impact On The Field Of Automl:**

This paper presents an interesting idea on how to optimize the feature extraction. The approach seems to be well thought through. Its current limitation to numerical original features would probably limit its impact, but on the other hand the authors wrote to tackle this issue in future work.

**Reproducibility:**

The general approach could probably be reproduced by the description of the paper, especially if some minor issues are addressed. I am wondering what happens if the randomly generated feature is already contained in the candidate feature set. Apart from that, the exact definition of the used ML methods, i.e. their parameter values, are not given in the paper.

Without trying to execute the code provided via 4open.sience, I doubt that it can be simply executed due to specific configurations of a HPC as mentioned in the paper. Unfortunately, the readme does not provide information about how to use DIFER without their HPC execution scripts, i.e. installing it locally and letting it run directly via python.

**Review Confidence:**

4: You are confident in your assessment, but not absolutely certain. It is unlikely, but not impossible, that you did not understand some parts of the submission or that you are unfamiliar with some pieces of related work.

**Review Rating:**

5: Accept, good paper

**Review Summary:**

Overall, the idea of the presented paper is very interesting. Unfortunately, the descriptions are often very short and for someone who did not see such a usage of an encoder decoder before, hard to understand. The images are missing important information, the ordering of the figures and tables is not reasonable. The evaluation is a good foundation, but the some experiment setups do not provide a fair comparison or do not cover all aspects or datasets. All in all, this paper seems to be not carefully finalized.

The final decision will be dependent on the improvements on the methodology and evaluation (Sec. 3 and Sec. 4). If the concerns are tackled, I am willing to fully accept the paper.


---
# Update after Rebuttal Phase

The authors addressed most of my concerns. Especially, changes have been made in Sec. 3 (Problem Formulation) and Sec. 4 (Experiments), which improved the correctness and clarity a lot. A section giving an overview of DIFER has been added, which helps to understand the overall approach, before going into details. The overview figure has been modified to correctly contain all key aspects of the approach. Some experiments have been added or modified to have a better evaluation of the approach and a better understanding of its results.

Nevertheless, some descriptions or discussions are quite short and limited. Some experimental results are mentioned in discussions without having the results explicitly given in the paper. Further usage of the appendix would be suitable here.

**Technical Quality And Correctness:**

Although the presented idea seems to be interesting, the way it is formally described and evaluated has a lot of improvement potential (especially Sec. 3 and Se. 4)).

The formal definition is not well defined. E.g. f_i is used for the i-th raw features and for the newly constructed feature, which should be separated to prevent irritation, e.g. a simple f could be used in case of a newly constructed feature. Additionally, I am wondering if it is the goal of AutoFE in general to return a set of features, that always contains the original features, I would have expected the final set can, but must not, contain all the original features. Furthermore, some definitions of the variables are not well defined (like E). The neighborhood of the feature optimization (Eq. 5) seems to be missing a set definition.

In the experiment setup, additional information about the implementation would be helpful, like which libraries are used (e.g. for the ML algorithms) etc. I am wondering what the search space of the other methods are, as the comparison should also be made on the same search space, so that the improvement on high-order functions are clearly separable. Tbl. 1 shows the size of the returned features, but only for DIFER, so there is no comparison in the numbers of features possible.

The efficiency evaluation with the comparison of number of feature evaluations (RQ 2), i.e. pipeline executions, is interesting, but not complete, as the training time of the score predictor should also be considered. Here, e.g. runtime would be an additionally informative metric.

The evaluation of the effectiveness of high-order features (RQ 3) is not well done. Features with order > k/2 (as k is 5, this means features with order 4 or 5) are drawn in the same color and represent the majority of features for almost all datasets, and therefore the authors conclude that DIFER is effective wrt high-order features, although it is completely unclear if those high-order features are just of order 4. Here, a color for each order would provide more insight in the amount of different order features. Furthermore, there are only 23 of the 25 dataset depicted in Fig. 4a. The way the 5 representative datasets are chosen is not further described, nor is a differentiation in classification or regression given. Actually, there are 4 regression datasets and only one classification dataset. I am wondering why this comparison is not done for all datasets.

The evaluation of different ML algorithms (RQ 4) is missing the baseline with Random Forest in Tbl. 2. The presented avg and min/max improvement would benefit noting the standard deviation.

Another evaluation (suggested RQ 5) could investigate the performance/score of the predictor in terms of the correctness of the predicted performance. Here it might be suitable to not compare the absolute performance, but the ranking that can be generated using the predictions.

In general, information about the consumed resources, i.e. CPU/GPU hours are missing.

---

### Meta-Review · Area_Chair_Nn7y · 2022-05-09

**Recommendation:** Accept
**Confidence:** 4

**Metareview:**

The reviewers have been a bit divided on this paper, and there was plenty of discussion. I do think that the paper could use some improvement, specifically about one concern risen on the explanation *why* the method works, which is also a concern to me. However, the topic is interesting, and reviewers expect some impact of this work.

In accordance with the reviewers, I consider the paper borderline. It is fair (and I would suggest) to accept it, but in its current shape it is clearly not a must-have for the conference.

---

### Decision · Program_Chairs · 2022-05-13

Accept